# Mechanically Enhanced Electrical Conductivity of Polydimethylsiloxane-Based Composites by a Hot Embossing Process

**DOI:** 10.3390/polym11010056

**Published:** 2019-01-02

**Authors:** Xiaolong Gao, Yao Huang, Xiaoxiang He, Xiaojing Fan, Ying Liu, Hong Xu, Daming Wu, Chaoying Wan

**Affiliations:** 1College of Mechanical and Electrical Engineering, Beijing University of Chemical Technology, Beijing 100029, China; gaoxiaolong@mail.buct.edu.cn (X.G.); fxj0803@126.com (X.F.); liuying@mail.buct.edu.cn (Y.L.); xuhong@mail.buct.edu.cn (H.X.); 2State Key Laboratory of Organic-Inorganic Composites, Beijing University of Chemical Technology, Beijing 100029, China; hy06@163.com; 3International Institute for Nanocomposites Manufacturing (IINM), WMG, University of Warwick, Warwich CV4 7AL, UK

**Keywords:** electrical conducting network, forced assembly, compression-induced percolation threshold, hybrid filler, synergy

## Abstract

Electrically conductive polymer composites are in high demand for modern technologies, however, the intrinsic brittleness of conducting conjugated polymers and the moderate electrical conductivity of engineering polymer/carbon composites have highly constrained their applications. In this work, super high electrical conductive polymer composites were produced by a novel hot embossing design. The polydimethylsiloxane (PDMS) composites containing short carbon fiber (SCF) exhibited an electrical percolation threshold at 0.45 wt % and reached a saturated electrical conductivity of 49 S/m at 8 wt % of SCF. When reducing the sample thickness from 1.0 to 0.1 mm by the hot embossing process, a compression-induced percolation threshold occurred at 0.3 wt %, while the electrical conductivity was further enhanced to 378 S/m at 8 wt % SCF. Furthermore, the addition of a second nanofiller of 1 wt %, such as carbon nanotube or conducting carbon black, further increased the electrical conductivity of the PDMS/SCF (8 wt %) composites to 909 S/m and 657 S/m, respectively. The synergy of the densified conducting filler network by the mechanical compression and the hierarchical micro-/nano-scale filler approach has realized super high electrically conductive, yet mechanically flexible, polymer composites for modern flexible electronics applications.

## 1. Introduction

The development and popularity of smart electronics have attracted increasing demands for high performance and mechanically flexible electrically conducting polymer composites. Intrinsic electrical conducting polymers, such as polypyrrole, polyaniline, polyacetylene, and polythiophene, have the constraints of unstable conductivity, insolubility, difficulty of processing, small-scale, and high cost. Electrically conducting polymer composites can be prepared by incorporating conductive fillers into an insulating polymer matrix and provides the advantages of ease of processing and an ability to tailor the electrical properties based on composite architecture [1,2,3,4,5,6]. The electrical conductivity of polymer composites relies on the nature of the conductive fillers and the formation of a continuous and percolated conducting pathway in the insulating polymeric matrix [7,8,9,10,11]. The dispersion of conducting fillers such as carbon nanotubes (CNTs), carbon fibers, graphene, and carbon black, in a polymer matrix have been improved by using in situ polymerization [12] or forming segregated microstructures under specific processing conditions [13]. It has been reported that multi-walled CNTs formed a segregated and percolated conductive network structure in poly (phenylene sulfide) matrix by using solid-mixing and subsequent melt-compression. A percolation threshold as low as 0.33 wt % of CNTs was obtained, and the electrical conductivity of the composites was increased from ∼10^−10^ S/cm to ∼0.11 S/cm with the use of CNT contents up to 10 wt %.

The electrical percolation threshold and electric conductivity of polymer composites are highly dependent on the properties of the conducting fillers, their dispersion, and interfacial interactions between the filler particles and the polymer matrix [14,15,16,17,18,19]. A good dispersion of graphene oxide (GO) was achieved in polyamide-6 (PA6) composites by in situ polymerization, which facilitated the formation of an electrically conductive network and resulted in significantly improved electrical conductivity of the composite [12]. A PA6/polypropylene (PP)/CNT (20/80/4) composite exhibited a high electrical conductivity in comparison with the PA6/PP/CNTs (50/50/4) composite due to the selectively localized CNT in the PA6 phase in forming a compact conductive network [20]. In addition, a synergistic effect between graphene and CNTs in a thermoplastic polyurethane matrix led to a reduced percolation threshold, where the graphene acted as a ‘spacer’ to separate the entangled CNTs from each other and the CNT bridged the gap between individual graphene sheets. Such an effect was beneficial for the dispersion of the CNTs and the formation of effective conductive paths, leading to improved electrical conductivity at a lower conductive filler content [21]. As to conductive flexible polydimethylsiloxane (PDMS) composites, Sethi prepared elastomeric composites by incorporating various carbon blacks in insulating PDMS matrix. The results showed that alternating current (AC) and direct current (DC) conductivity increased with an increase in some bending flex cycles, and with compressive flexing, DC conductivity initially dropped but later started increasing with an increase in flex cycles [22]. Moreover, to address the challenge of flexible materials with a relatively high electrical conductivity and a high elastic limit, Wu reported a new and facile method to prepare porous polydimethylsiloxane/carbon nanofiber composites by using sugar particles coated with carbon nanofibers (CNFs) as the templates. The resulting three-dimensional porous nanocomposites, with the CNFs embedded in the PDMS pore walls, exhibit a greatly increased failure strain (up to ∼94%) compared to that of the solid, neat PDMS (∼48%) [23]. More related research on flexible PDMS can be found in References [24,25,26,27,28].

The electrical conductivity of filled polymer composites generally reaches saturation and becomes independent of filler loading once the filler loading level is above the percolation threshold. As a result, the enhancement of the electrical conductivity of polymer composites is very limited [29,30,31,32]. To promote the maximum electrical conductivity of polymer composites, we proposed a spatial confining forced network assembly (SCFNA) method to increase the packing density of the conducting network in the composites by mechanical compression [8,33,34], which reduced the separation distance between the fillers by excluding the insulting polymer phase out of the network. This mechanical compression approach effectively enhanced the electrical conductivity of polypropylene/short carbon fiber (SCF) composites by 2–4 orders of magnitude [33] and thermal conductivity of PDMS/SCF composites by 7.79 times higher when the sample thickness was reduced to 0.3–0.5 mm [34]. In this work, we further investigate the effects of hybrid filler systems combined with the SCFNA method on the electrical conductivity of PDMS composites. The synergy between the micro- and nano-scale fillers are shown to enhance the continuity of conductive pathways in the forced assembled network. This method can facilitate the formation of a compact conducting network, which is applicable for enhancing the electrical and thermal conductivity, and mechanical toughening polymer composites.

## 2. Experimental

### 2.1. Materials

PDMS with a trade name of SYLGARD 184, obtained from DOW CORNING (Midland, MI, USA), was used as the polymer matrix. As a micro-scale filler; short carbon fibers (SCF) of 5–10 μm in diameter and 3–5 mm in length were provided by Toray, Japan. As nano-scale filler; graphene, carbon black, and CNTs were used. Few-layer graphene (G) of 1.0–1.77 nm in thickness and 10–50 μm in width was provided by Changzhou Sixth Element Material Technology Co., Ltd. (Changzhou, China). Superconductive carbon black (CCB, BP2000) with a particle size of 15 nm was obtained from CABOT (Boston, MA, USA). CNTs with 20–30 nm diameter and 10–30 μm length were provided by Beijing Daoking Technology Co., Ltd. (Beijing, China). The electrical conductivity of the four fillers (SCF, G, CCB, and CNT) were ~ 2.6 × 10^4^ S/m, 1.0 × 10^5^ S/m, 5.0 × 10^4^ S/m, and 1.0 × 10^4^ S/m, respectively.

### 2.2. Composite Preparation and SCFNA Process

A conical twin-screw compounder of HAAKE MiniLab combined with a specifically designed hot embossing device was used for preparation of the PDMS composites in order to realize a spatially confined forced assembly network (SCFNA) of conductive fillers. 

For PDMS/SCF composites, the PDMS was mixed with the short carbon fibers at weight ratios of 0.5, 1, 1.5, 2, 3, 4, 6, or 8 wt % at 25 °C with a rotational speed of 70 rpm for 10 min. For the preparation of PDMS composites containing hybrid micro- and nano-scale fillers, 1 wt % of CNT, G, or CCB was included to the above PDMS/SCF compound with a rotational speed of 50 rpm for an additional 10 min at 25 °C to obtain PDMS/SCF-CNT, PDMS/SCF-G, or PDMS/SCF-CCB composites. 

The experimental hot embossing device as shown in Scheme 1a provided a pressing force up to 10 MPa with a pressing speed of 0.005 to 0.5 mm/s in a working stroke. The servo motor of this embossing device was accurately controlled by a programmable logic controller (PLC) system and the position accuracy of ±3 μm for the up platen was guaranteed. The compressing mold was made of two flat plates with electrical heaters. The temperature of the molds was controlled from ambient temperature to 350 °C with a control accuracy of ±1 °C and heating rate of 15 °C/min. The effective working area was 210 mm × 300 mm.

The compounded materials were compressed in the designed mold at 20 °C following the steps shown in Scheme 1b. The compounds were compressed from an original thickness of 10 to 1.0 mm within 120 s and kept at 10 MPa for 5 min to initiate the first assembly of the filler network. Then the composite sheets were continuously compressed to the thickness of 0.4, 0.3, 0.2, and 0.1 mm, respectively within 30 s. The second compression aims to force the assembly of the filler network in a spatially confined space. In this work, when the sample thickness (H) was reduced below a critical network thickness (*d_m_* = 0.3 mm), the conductivity was significantly enhanced. Finally, the composite sheets were cured at 100 °C for 10 min to stabilize the filler network. The specimens with a thickness of 1.0 mm after the first stage of compression were used as a control.

### 2.3. Characterization

A video measuring system JTVMS-1510T manufactured by Dongguan JATEN Precision Instrument Co., LTD. (Dongguan, China) was used to investigate the morphology evolution process during the preparation of the compact short carbon fiber network in the PDMS matrix. A Hitachi S4700 scanning electron microscopy (SEM, Tokyo, Japan) was used to study the dispersion and localization of the fillers in the composites under an accelerating voltage of 30 kV.

The electrical conductivity of the composite samples was measured using a Keithley 4200-SCS (Cleveland, OH, USA) with a standard four-probe method and a ZC-90D resistivity meter from Shanghai Taiou Electronics (Shanghai, China). All electrical conductivity measurements were carried out at ambient conditions.

The mechanical properties of the PDMS composites were evaluated using a universal material-testing machine (UTM-1422, Chengde Jinjian Testing Instrument Co., Ltd., Chengde, China) with a cross-head speed of 5 mm/min at ambient temperature and the corresponding dimension of the specimen is 30 mm × 15 mm × 0.2 mm.

## 3. Results and Discussion

### 3.1. Concentration-Induced Electrical Percolation of PDMS Composites

The DC electrical conductivity of the PDMS composites containing short carbon fibers was investigated as a function of filler concentration or mechanical compression, and the effects of an additional nano-scale filler (CNT, G, or CCB) on the electrical properties of the composites were also discussed. The nano-scale filler was added at a loading level of 1 wt % and the sample thickness of the PDMS composites was reduced from the original 1.0 mm further down to 0.1 mm using the SCFNA compression method.

The electrical conducting behavior of polymer composites can be interpreted by the classic percolation theory [35]. The electrical conductivity of the composites near the percolation threshold follows a power-law relationship described in Equation (1),
(1)$$\sigma \propto \sigma_{0}{\left(\varphi -\varphi_{c}\right)}^{n},\;\mathrm{for}\;\varphi \geq \varphi_{c}$$
where *σ* is the electrical conductivity of composites, *σ*_0_ is a constant that is typically assigned to the plateau conductivity of fully loaded composites, *φ* is the filler weight content, *φ*_c_ is the weight ratio of the filler at the percolation threshold, and the value of critical exponent (*n*) depends on the system dimension and is used to interpret the mechanism of network formation. An insulator-conductor transition is observed at the *φ*_c_, and the electrical conductivity of the polymer composites increases with the filler content increasing when $\varphi \geq \varphi_{c}.$ As shown in Figure 1a and Table 1, the DC electrical conductivity of the PDMS/SCF composites with a thickness of 1.0 mm was increased with an increase of SCF contents, from the original filler free PDMS of ∼10^−12^ S/m to 49 S/m when the SCF content was increased to 8 wt %. The *φ_c_* of PDMS/SCF composites was calculated to be 0.45 wt % and *n* = 7.45 according to Equation (1) [36,37,38], as shown in Figure 1b.

### 3.2. Compression-Induced Electrical Percolation of PDMS Composites

Similar to the concentration-induced percolation, a compression-induced percolation transition was observed during the SCFNA compression process, when the sample thickness or volumetric strain reached a critical value *ε*_c_ or under a critical pressure *P_c_* [39], as expressed in Equations (2) and (3):(2)$$\varphi_{c}=\frac{\varphi_{0}}{1-\Delta P_{c}}$$
(3)$$\varepsilon_{c}=1-\left(\frac{\varphi_{0}}{\varphi_{c}}\right)$$
where $\Delta P_{c}$ is the pressure increment (*P* − *P_c_*) to drive the percolation transition from $\varphi_{0}$ to $\varphi_{c}$. 

The percolation transition happened when the conducting fillers were approaching each other until a critical partial volume of the fillers was reached. As shown in Figure 1, when the thickness of the PDMS/SCF composites was reduced from 1.0 to 0.1 mm by the SCFNA compression, the overall conductivity of composites was increased significantly and the *φ*_c_ was reduced to 0.3 wt %. Above the percolation threshold, the increase of the short carbon fiber concentration from 6 to 8 wt % led to a small increase of the ultimate conductivity of the PDMS/SCF composite (1 mm) from 44.6 to 48.6 S/m, only increasing by 8.97%. However, for the PDMS/SCF composite samples (0.1 mm) prepared by the SCFNA method, the conductivity was increased by 47.1%, from 257 to 378 S/m. This illustrates that when a conducting network is formed in the polymer matrix (above the concentration induced percolation threshold), a further increase in filler concentration has little contribution to the ultimate electrical conductivity. In contrast, compression may densify the conducting network, and further enhance the electrical conductivity without increasing filler concentrations. This is particularly promising in the case that the ultimate electrical conductivity of polymer composites above the concentration-induced percolation threshold can be further enhanced by a compression-induced percolation transition. 

Figure 2a shows the effects of sample thickness on the electrical conductivity of the PDMS/SCF composites at above its percolation threshold. As the sample thickness was reduced, the electrical conductivity of the PDMS/SCF composites was slowly increased and exhibited an abrupt enhancement when the thickness was below *d_m_* = 0.3 mm, which indicates that a new compression-induced percolation transition occurs, i.e., a compact assembly of filler network at a higher compression ratio. A further reduction of the sample thickness to 0.1 mm led to a significant increase of the conductivity, from the original 28 S/m (at 1 mm) up to 187 S/cm for PDMS/SCF (4 wt %). Similarly, the PDMS/SCF (4 wt %)-CNT (1 wt %) composite had the electrical conductivity increased from 65 to 511 S/m. As shown in Figure 1b, for the compressed composites with 0.1 mm in thickness, the addition of 1 wt % of a second nano-scale filler such as CNT or CCB reduced the *φ*_c_ from 0.3 to 0.15 wt %, demonstrating a synergistic effect between the micro-scale short carbon fibers and the second nano-scale filler.

The effects of a second nano-scale filler along with the compression ratio (R) on the electrical conductivity of the composites are compared in Figure 2b. The compression ratio R is defined as 1.0 mm over the assembly thickness of composites prepared by SCFNA, i.e., when the thickness of the composite samples was reduced to 1, 0.4, 0.3, 0.2, and 0.1 mm, the compression ratio R was 1, 2.5, 3.3, 5, and 10, respectively. 

As shown in Figure 2b, the conductivity of the composites is generally increased with an increase of the compression ratio R of the composites. The conductivity of PDMS/SCF, PDMS/SCF-G, PDMS/SCF-CNTs, and PDMS/SCF-CCB were enhanced by 6.68, 6.52, 7.41, and 7.86 times after simply compression-molding of the composite sheets from 1 to 0.1 mm. This indicates that the thickness or the packing density of the filler network plays a significant role in the electrical properties of the composites. The addition of 1 wt % of graphene to the PDMS/SCF composites did not show any obvious effect on the conductivity under variable compression ratios, as shown in Figure 2a,b, while the use of 1 wt % CCB and CNTs significantly increased the conductivity along with the thickness reduction. The CNTs were the most effective, which can be ascribed to their large aspect ratio and bridging the conducting network easily compared with CCB.

To demonstrate the dependence of electrical conductivity on partial volume of the fillers (sample thickness), the strain-sensitivity coefficient *Ks* is plotted against the filler concentration according to Equation (4), and is shown in Figure 2c,d.
(4)$$K_{S}=\frac{d\sigma }{\sigma }/\frac{dV}{V}$$
where *dV/V* is the volumetric strain, causing a relative increment of the conductivity ($\frac{d\sigma }{\sigma }$). 

As shown in Figure 2c, the electrical conductivity or *Ks* is not sensitive to the sample thickness when the SCF content is lower than 0.15 wt %. As the SCF content increases, 0.25 wt % ˂ *φ* ˂0.5 wt %, the *lg Ks* increases abruptly and reaches a maximum peak at 0.5 wt %, close to its percolation transition as determined in Figure 1b, indicating the formation of a compression-induced conducting network. The *lg Ks* gradually decreases as the filler concentration increases, indicating that once a compact conducting network has formed, the conductivity becomes less sensitive to the sample thickness and filler concentration. 

In the case of PDMS/SCF-CNTs, the addition of 1 wt % of CNTs facilitates the formation of a conducting SCF network at lower concentrations due to the gap-filling effect. In the SCF concentration range of 0.15 wt % ˂ *φ* ˂ 1 wt %, the *lg Ks* increases abruptly and then drops as the SCF concentration increases. The *lg Ks* peak value is lower than that of the PDMS/SCF composite, which is because the addition of CNTs makes the network more compact, and thus reduces the strain sensitivity. 

As shown in Table 1, the electrical conductivity of PDMS/SCF-G, PDMS/SCF-CCB, and PDMS/SCF-CNTs composite samples (thickness is 0.1 mm) prepared by SCFNA increased significantly from 1.38 × 10^−5^ to 383 S/m, 1.21 × 10^−3^ to 657 S/m, and 6.51 × 10^−4^ to 909 S/m with an increase of the short carbon fiber content from 0 to 8 wt %, respectively. Therefore, the electrical conductivity of PDMS composites can be more effectively enhanced by constructing a compact conducting network instead of increasing the filler loading. By further increasing the packing density by adding a second nanofiller or increasing the filler concentration, the electrical conductivity can be significantly further enhanced. 

### 3.3. Characterization of the Conducting Filler Network in PDMS

Reducing the gap between adjacent filler particles would benefit electron conduction [8,40,41], since the electrons can transport between adjacent conducting fillers within an average distance less than 100 nm, according to conductive passage theory [42]. Figure 3 shows the morphology evolution of the short carbon fiber network during the preparation of the PDMS/SCF (4 wt %) composites. A homogeneously-dispersed short carbon fiber network was formed in the PDMS matrix after melt-compounding, as shown in Figure 3a. When further subjected to the strong shearing force in the conical twin-screw extrusion process, the short carbon fiber filler agglomerated and self-assembled, as shown in Figure 3b. After the first stage of compression, the self-assembly network was formed, as shown in Figure 3c, and was subsequently condensed by the second stage of spatial confining compression (SCFNA) to create a forced assembly network in Figure 3d. From the cross-sectional images in Figure 3e, the short carbon fiber framework was compacted by the forced assembly and the average distance between the short carbon fibers was reduced from more than 15 μm to less than 100 nm, which creates the electron conduction pathways through both field emission and conductive passage. Figure 4 shows the SEM images of the cross-section of PDMS/SCF-CNT samples. It is observed that CNTs and SCF are homogeneously dispersed in PDMS before compression, and most CNTs are dispersed in PDMS rather than around SCF, as shown in Figure 4a. After self-assembly, most CNTs aggregated to form a conductive network and moved towards SCF as shown in Figure 4b. Figure 4c and Figure 5b show that most CNTs were dispersed around SCF, filling the gap between the SCF network after forced compression. The more compact SCF network and CNT network help enhance the conductivity of PDMS/SCF-CNT composites as a result of a synergistic effect.

The gap-filling effect of an additional nano-scale filler on the conducting network of short carbon fibers is illustrated by SEM imaging in Figure 5. In the PDMS/SCF-CCB composites, the majority of the CCB particles are located inside the network of short carbon fibers which fill the space between short carbon fibers, as shown in Figure 5a. A synergistic effect of a combination of CCB and short carbon fibers could enhance the continuity of conductive pathways in the compact carbon fiber conductive forced assembled network. As shown in Figure 5b, most CNTs are dispersed regularly around the short carbon fiber framework. The average spacing of adjacent CNTs in the network is less than 100 nm, and the interconnection between the long CNTs with the compact short carbon fiber network accounts for the enhanced conductivity of PDMS/SCF-CNT composites. In comparison, the few-layer graphene sheets are mostly dispersed in the PDMS matrix rather than inside of the carbon fiber network, as shown in Figure 5c, implying a lower synergy efficiency of the combination of graphene and SCF in PDMS composites. In addition, the agglomerates of graphene sheets may also account for the low conductivity and poor contribution for the PDMS/SCF composites.

Ohm’s Law can be used to explain why the reduced separation between filler particles of the conductive network could effectively improve the electrical conductivity of composites. As seen in Figure 5d, when the adjacent short carbon fibers are closer, the effective resistance of the PDMS region between the short carbon fibers decreases, as a result, the electrical conductivity of the PDMS/SCF composites is increased. Moreover, with the addition of a second nano-scale filler, a gap-filling conductive network is formed among the SCFs, as shown in Figure 5e of the conductive model for PDMS/SCF-CNT and PDMS/SCF-CCB composites. This indicates that the PDMS resistance between adjacent SCFs is parallel with a small resistance, which greatly improves the conductivity of the composites.

Figure 6 shows the mechanism of the SCFNA process and the structure development of conducting a filler network in polymer composites. Figure 3 shows the mechanism of the morphology evolution of shorter carbon fiber network in PDMS. The SCF was homogeneously dispersed in PDMS by melt-compounding, as shown in Figure 3a, due to the strong shearing and stretching stress in the conical twin-screw extrusion process. During the two-step compression, when the thickness of the sample was larger than *d_m_*, the SCF could assemble to form a conducting network which could wiggle in the liquid PDMS by free compression. This caused the self-assembled SCF network to become more compact, as shown in Figure 3c. At the following forced assembly process, namely, forced compression, the sample thickness was further reduced to below *d_m_*; a noteworthy densification of the network threads was therefore formed. Due to the restriction of wiggling freedom of the network, the SCF at the threads would tend to get closer which squeezed the extra polymer out of the filler granules in a special confining compression process, as shown in Figure 3d. The compression force could be transmitted to the granules to break the balance of the self-assembly network, and the self-assembled network was forced to pack more densely, which resulted in enhanced electrical conductivity [33]. 

Therefore, the SCFNA process provides a simple route to construct a continuous and compact short carbon fiber micro-network in the polymeric matrix by simply applying mechanical force, which enhances the electrical conductivity up to 378 S/m, 7.8 times higher than that of the PDMS/SCF composites by compounding. Moreover, the gap-filling effect of a second nano-scale filler such as CNTs has interconnected the micro-SCF network. As a result, the electrical conductivity of the PDMS composites containing hybrid fillers and processed via SCFNA can be further enhanced to as high as 909 S/m. 

### 3.4. Mechanical Properties

The mechanical properties of PDMS composites with SCF (0–8 wt %), and 1 wt % of a nano-sized filler (CNT, CCB, and G) are shown in Table 2—the sample thickness is 0.2 mm. The maximum tensile strength of PDMS/SCF composites decreased from 4.5 ± 0.1 MPa to 2.4 ± 0.1 MPa when the SCF content increased from 0 to 8 wt %. The reduced properties can be ascribed to the higher contents of micro-SCF and the poor interfacial interaction between SCF and PDMS. After addition of 1 wt % nanofillers into PDMS/SCF (4 wt %) composites, the tensile strength and elongation at break of the composites were enhanced. Graphene showed a better mechanical reinforcement effect than CNT and CCB. With the addition of 1 wt % graphene, the maximum tensile strength of PDMS/SCF composites improved from 2.8 ± 0.1 MPa to 3.8 ± 0.2 MPa. The two-dimensional graphene nanosheets offer a stronger interface layer with PDMS as compared to one-dimensional CNT or granular CCB, and the graphene is mainly located in the PDMS rich regions of the composites, which would be effective in reinforcement of PDMS. Meanwhile, graphene presents a folded morphological distribution in PDMS as shown in Figure 5c, which would benefit the enhancement of the tensile strength and elongation at break. Nevertheless, both the CNT and CCB improved the tensile strength of PDMS/SCF composites, because the nanosized CNT and CCB interact with the PDMS matrix stronger than the microsized SCF. Figure 7 shows the tensile curve of PDMS/SCF (8 wt %) and PDMS/SCF (8 wt %)-CNT (1 wt %) samples. 

## 4. Conclusions

The incorporation of conducting fillers into an insulating polymer matrix can generally enhance the electrical conductivity to a limited extent when beyond the percolation threshold. In this work, a spatially confined forced network combined with a hybrid micro-nanoscale filler approach were investigated in order to prompt highly electrical conductive polymer composites. With our newly-designed spatially confined forced network (SCFNA) process, the electrical conductivity of PDMS/short carbon fiber (SCF) composites was increased by 7.8 times in comparison to the traditional compounding method. The addition of a nano-scale filler, such as CCB or CNT, further increased the conductivity over 2 times higher at above the percolation threshold of SCF; this was a result of a synergy effect of the hierarchical filler network. The spatially confined forced network provides a simple route to construct a continuous and compact conducting filler network in PDMS/SCF composites and results in super high electrical conductivity yet mechanical flexible composites with lower filler concentrations. This approach will significantly prompt the applications of conductive polymer composites in the areas of flexible sensors and intelligent wearable products.

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
