# Peer review of "Mechanically Enhanced Electrical Conductivity of Polydimethylsiloxane-Based Composites by a Hot Embossing Process"

_polymers, 2019, doi:10.3390/polym11010056_

Round 1

Reviewer 1 Report

The manuscript entitled “Mechanically Enhanced Electrical Conductivity of Polydimethylsiloxane-Based Composites by a Hot Embossing Process. This work is of great interest to the flexible electronics applications. Nonetheless, the manuscript can and should be further improved by taking the following aspects into consideration.

1.  Flexibility test is missing? It is important to include DC electrical conductivity of PDMS based composites under flat and bends condition.       

2. It is important to include the tensile curve of PDMS/8 wt% SCF samples with layers of insulating surface.

3. Introduction: It is important to include the scope/limitations of PDMS based composites and also other flexible electronics devices for different applications in detail. Please cite some relevant references: Polym. Int, 66: 1295-1305, ACS Appl. Mater. Interfaces 2017, 9, 16, 14207-14215, Journal of the mechanical behavior of biomedical materials 61 (2016) 87–95, Bioelectronic Medicine 2018, 4:6., Sensors and Actuators B: Chemical 200, 227-234, Adv. Energy Mater. 2018, 8, 1701791.  

4. Please check and correct the font size of x and y-axis of all figures in the manuscript. It should be the same.

5. There are some grammatical and syntax error, please check and correct

6. The quality of some figures is very poor and needs to be enhanced.

Author Response

Re: Manuscript ID: polymers-408982

Dear Reviewer,

Many thanks for all the comments. We have carefully revised the manuscript and addressed all the comments point-by-point. The revisions were highlighted in red in the manuscript and also listed below. 

Reviewer 2 Report

In this paper, authors reported the production of super high electrical conductive polymer composites by a hot embossing design. With the addition of SCF and nanofillers, the conductivity of PDMS was increased to 909 S/m. Authors attributed the remarkable increase of the conductivity to a synergy effect of the formation of PDMS/SCF/nanofiller hierarchical filler network. Some comments are as follows.

 Comments:

1.     Authors should discuss the mechanism of the morphology evolution of shorter carbon fiber network in PDMS as displayed in figure 3.  

2.     Authors measured the morphology evolution of PDMS/SCF composites. However, the PDMS/SCF/nanofiller showed the best performance. It is obvious that the analysis and discussion the morphology evolution of PDMS/SCF/nanofiller composites is more important. The reviewer strongly recommends authors to do this.

3.     In Line 311, the authors wrote “Nevertheless, both the CNT and CCB improved the tensile strength of PDMS/SCF composites.” Authors should explain the reason.

Author Response

(The authors gave the same response as above.)

Reviewer 3 Report

This manuscript introduces super high electrical conductive polydimethylsiloxane (PDMS) /carbon nanomaterials composites fabricated by hot embossing machine, which provided both a pressing force up to 10 MPa with a pressing speed of 0.005 to 0.5 mm/s and heat controlled to 350 by electrical heater. When reduced the sample thickness from 1.0 mm to 0.1 mm by the hot embossing process, the electrical conductivity was enhaned from 48.6 S/m to 378 S/m at 8 wt% short carbon fiber (SCF). In addition, the additional of a second nanofiller of 1 wt%, such as carbon nanotube or conducting carbon black further increased the electrical conductivity of the PDMS/SCF (8 wt%) composites to 909 S/m and 657 S/m, respectively. However, the authors should considerably improve the manuscript before consideration for publication. The following issues should be considered in order to strengthen this manuscript.

1) When the conductive polydimethylsiloxane (PDMS) composite which contained short carbon fiber (SCF), I cannot find the difference of instrument and experimental method between the hot embossing process used in this experiment and two-stage compressing used in previous research (RSC Adv., 2017, 7, 14761–14768). You need to emphasize what is a difference or  improvement

2) Mechanical properties of PDMS/SCF (8 wt%) composites as well as PDMS/SCF (4 wt%) composites should be presented because the optimized condition of the conductivity were PDMS/SCF (8 wt%)

3). In addition, please add references: Scientific Reports, 2018, 8, 12313; Nanoscale, 2017, 9, 5072-5084; Scientific Reports 2018, 8, 1375

Author Response

(The authors gave the same response as above.)
